# Prevalence and Severity of Pelvic Floor Disorders during Pregnancy: Does the Trimester Make a Difference?

**DOI:** 10.3390/healthcare11081096

**Published:** 2023-04-11

**Authors:** Yoav Baruch, Stefano Manodoro, Marta Barba, Alice Cola, Ilaria Re, Matteo Frigerio

**Affiliations:** 1Urogynecology and Pelvic Floor Unit, Department of Obstetrics and Gynecology, Tel Aviv Medical Center, Tel Aviv University, Tel Aviv 6997801, Israel; 2Department of Obstetrics and Gynecology, ASST Santi Paolo e Carlo, San Paolo Hospital, 20132 Milano, Italy; 3Department of Obstetrics and Gynecology, Fondazione IRCCS San Gerardo dei Tintori, University Milano Bicocca, 20900 Monza, Italy; m.barba8792@gmail.com (M.B.);

**Keywords:** PFQPP, pregnancy, pelvic floor disorders, sexual dysfunction, quality of life, stress urinary incontinence, pelvic organ prolapse, urge urinary incontinence

## Abstract

(1) Background: Women experience pelvic floor dysfunction symptoms during pregnancy. This study is the first to investigate and compare variances in the prevalence and severity of pelvic floor symptoms between trimesters using a valid pregnancy-targeted questionnaire. (2) Methods: A retrospective cohort study was conducted between August 2020 to January 2021 at two university-affiliated tertiary medical centers. Pregnant women (*n* = 306) anonymously completed the Pelvic Floor Questionnaire for Pregnancy and Postpartum with its four domains (bladder, bowel, prolapse, and sexual). (3) Results: Thirty-six women (11.7%) were in the 1st trimester, eighty-three (27.1%) were in the 2nd trimester, and one hundred and eighty-seven (61.1%) were in the 3rd trimester. The groups were similar in age, pregestational weight, and smoking habits. A total of 104 (34%) had bladder dysfunction, 112 (36.3%) had bowel dysfunction, and 132 (40.4%) reported sexual inactivity and/or sexual dysfunction. Least prevalent (33/306; 10.8%) were prolapse symptoms. Increased awareness of prolapse and significantly higher rates of nocturia and the need to use pads due to incontinence were recorded in the 3rd trimester. Sexual dysfunction or abstinence were equally distributed in all three trimesters. (4) Conclusions: Bladder and prolapse symptoms, equally frequent throughout pregnancy, significantly intensified in the 3rd trimester. Bowel and sexual symptoms, equally frequent throughout pregnancy, did not intensify in the third trimester.

## 1. Introduction

Pelvic floor dysfunction (PFD) is a common multifactorial heterogeneous condition mainly consequent to weakening of the pelvic floor following vaginal or operative delivery. Otherwise, PFD can be related to genetic, structural and or hormonal factors [1]. PFD symptoms include vaginal, bowel, lower urinary tract, and sexual impairments such as urinary and anal incontinence, overactive bladder, pelvic organ prolapse, and sexual discomfort. PFD symptoms are widespread and are known to affect millions of women worldwide. Almost a quarter of women in the United States suffer from at least one pelvic floor disorder whether urinary incontinence, fecal incontinence, or pelvic organ prolapse [2]. It remains difficult, however, to determine the true prevalence of PFD because many women who suffer from FFD symptoms choose not to seek medical assistance and some are even reluctant to discuss PFD symptoms with their caregivers. The prevalence of PFD increases with older age and elevated body mass index [3,4]. Other than age and obesity, risk factors for PFD include smoking, mode of delivery, familial predisposition, race, and connective tissue disorders [5]. It has been shown that familial cases tend to portray more than one pelvic floor defect suggesting that underlying genetic factors may enhance PFD morbidity [6]. PFD negatively affects the social and physical functions of women, restricts their daily activities, impairs sexual function, and ultimately reduces their overall quality of life (QoL) [7]. The widespread prevalence of PFD symptoms, although far from being life threatening, results in lost productivity and a considerable economic burden on healthcare resources.

Women experience an increase in urinary incontinence, pelvic floor organ prolapse, sexual handicaps, and colorectal symptoms during pregnancy [8,9]. Involuntary loss of urine upon sneezing or coughing in pregnant women is detrimental to the quality of life. The quality of life of women with PFD symptoms is expected to decrease with increasing gestational age.

The use of validated QoL questionnaires rather than clinical interviews is effective and useful for the assessment of PFD symptoms as regards to their presence, severity, and their impact on patients’ quality of life, bringing to light conditions that may otherwise remain unrevealed [10]. The German “Pelvic Floor Questionnaire for pregnant and postpartum women” (PFQPP) is a self-completed questionnaire that was recently constructed to cover all four essential domains of pelvic floor function (bladder, bowel, prolapse, and sexual function) and to assess PFD symptoms’ severity, their prevalence, and their impact on quality of life [11]. The questionnaire, which differentiates between women who report bothersome symptoms and those who do not, is a reliable and valid tool that incorporates 42 recognized items, and its original German version can be downloaded using the link in the Appendix A section [11]. The questionnaire integrates five ascertained risk factors: age over 35 years, familial predisposition, body mass index > 25 kg/m^2^, cigarette smoking, and a subjective inability to voluntarily contract the pelvic floor muscles [11,12].

The PFQPP translated into and validated in the Italian language [13] was recently submitted to 1048 pregnant women predominantly in the 3rd trimester (*n* = 927) recruited at eight hospitals in Italy. Almost half of the pregnant women suffered from PFD symptoms, one-third abstained from sexual activity, and half of them suffered from dyspareunia [14].

Pelvic floor distress symptoms have been adequately investigated in the third trimester of pregnancy whereas their presence in early pregnancy has been rather overlooked. We hypothesized that pelvic distress remains underreported in early and mid-pregnancy until symptoms worsen in the third trimester. Our objective was to delineate variances in PFD symptomatology between the three trimesters of pregnancy. We thus compared the prevalence and severity of PFD symptoms in a subset of women who anonymously completed the Italian PFQPP at different stages of pregnancy.

## 2. Materials and Methods

This study relays a secondary analysis from a cross-sectional study that was originally conducted in eight hospitals in Italy and Italian-speaking Switzerland [8].

The research protocol was approved by the local Institutional Review Board (*n*. 3116/2019) before the study began. Two tertiary medical centers namely, Santi Paolo e Carlo, San Paolo Hospital, Milan, Italy, and San Gerardo Hospital, Monza, Italy, who recruited women in all three trimesters, undertook this analysis. The PFQPP (validated in the Italian language), completed by 306 pregnant women, was used to evaluate and compare pelvic floor distress symptoms at different stages of pregnancy. The investigators distributed and collected the surveys from August 2020 to January 2021. Routine statistical methods were used to interpret the results.

Women with singleton pregnancies, 18 years and older, fluent in the Italian language, and recruited at antenatal outpatient clinics during routine pregnancy visits were asked to anonymously complete the Italian PFQPP. The exclusion criteria included: insufficient Italian language proficiency, diabetes mellitus, neurological disorders, and any other substantial co-morbidity. The women were subdivided between the pregnancy trimesters. Prevalence, severity, and risk factors were evaluated using the PFQPP.

### 2.1. The Pelvic Floor Questionnaire for Pregnancy and Postpartum—PFQPP

The Italian Pelvic Floor Questionnaire for Pregnancy and Postpartum embraces four domains of pelvic floor function (bladder, bowel, prolapse, and sexual function). The PFQPP weighs PFD symptoms’ severity and their impact on quality of life during pregnancy and postpartum. The patients were asked to consider how much their bowel, bladder, prolapse, and sexual issues bothered them. Responses that range from “not at all”; “a little”; “quite a lot”; and “very much” were deemed applicable. “I don’t have any symptom” was deemed not applicable. The total score in each domain defined the severity of PFD. The prevalence was calculated by the presence of bothersome symptoms when at least one point was granted.

The following items in the PFQPP were selected to assess the prevalence of specific PFD symptoms: Bladder question 5 was used to assess the prevalence of urge urinary incontinence re: “Do you leak urine before reaching the toilet when you have the desire to urinate?”. Bladder question 6 was used to assess the prevalence of stress incontinence re: “Do you leak urine when you laugh, sneeze, cough, or engage in sports?”. Bladder question 9 was used to assess the prevalence of incomplete bladder voiding re: “Do you feel like your bladder hasn’t completely emptied?”. Bowel question 4 was used to assess the prevalence of constipation re: “Do you experience constipation?”. Bowel question 8 was used to assess the prevalence of fecal incontinence re: “Do you have trouble maintaining stools or do you experience fecal leaks?”. Prolapse question 1 was used to assess the prevalence of prolapse symptoms re: “Do you experience a bulging sensation in the vagina?”. Sexual question 5 was used to assess the prevalence of dyspareunia: “Do you feel pain during intercourse?”. The magnitude of related symptoms was scored on a 5-point Likert Scale with the following choice of answers: “0 = not applicable—I do not have symptoms”, “1—not at all”; “2—a little”; “3—quite a lot” and “4—very much”.

The relationship between PFD symptoms and known risk factors including age over 35 years, familial predisposition (a family member with PFD symptoms), body mass index > 25 kg/m^2^, cigarette smoking, and a subjective inability to voluntarily contract the pelvic floor muscles was examined. Pelvic floor contraction (PFC) inability outlines the “subjective ability to voluntarily contract the pelvic floor muscles”. The question posed by the PFQPP “Are you capable to contract the muscles in the pelvic floor?” was used to assess PFC inability and the answers proposed are “yes”, “I do not know” and “no” [10]. The answer “no” designates pelvic floor contraction (PFC) inability.

### 2.2. Statistical Analysis

Categorical variables were summarized as frequency and percentage. Normally distributed continuous variables were reported as mean and standard deviation (SD) while skewed variables were reported as median and interquartile range. The Chi-square test and Fisher’s exact tests were applied to compare categorical variables between the three trimesters of pregnancy, while ANOVA and Kruskal Wallis tests were used to compare continuous variables between the three trimesters. All statistical tests were two-sided and a *p*-value of less than 0.05 was considered statistically significant. The Statistical Package for the Social Sciences (IBM SPSS software for windows), version 28, IBM Corporation, Armonk, New York, NY, USA, 2021) was used for all statistical analyses.

## 3. Results

The study cohort included 306 women. Thirty-six (11.8%) women were in the 1st trimester of pregnancy, 83 (27.1%) were in 2nd trimester, and one hundred and eighty-seven (61.1%) were in the 3rd trimester of pregnancy. The mean age at recruitment was 32.6 ± 4.6 years. The groups were comparable in age at recruitment and did not differ as regards to age, pregestational weight, and cigarette smoking (Table 1).

All women who gave consent to participate in the study completed at least part of the questionnaire (none were left blank) and the rate of missing items was 1%. Bladder, bowel, and sexual symptoms were frequent (reported by 34%, 36.3%, and 36.4% of participants, respectively) whereas prolapse symptoms were reported by only 10.8% of participants (Table 2).

### 3.1. Sexual Domain

Fifty-five women (55/306; 18.0%) reported absent sexual activity. Motives for sexual inactivity were a lack of a partner (21/55; 38.2%); partner-related inactivity (health-related, lack of desire, or fear of harming pregnancy) (22/55; 40%); a lack of sexual desire (9/55; 16.4%); and unpleasant or painful intercourse (3/55; 5.4%). These were equally distributed between the trimesters (*p* = 0.302). Of the remaining women, 65 (65/251; 25.9%) reported sexual dysfunction defined by answers “2—a little,” “3—quite a lot,” and “4—very much” to the question “How much do your sexual symptoms bother you?”. Sexual dysfunction was equally distributed in the 1st, 2nd, and 3rd trimesters of pregnancy (*p* = 0.611). The total sexual domain scores were 1.04, 1.25, and 1.25 in the 1st, 2nd, and 3rd trimesters, respectively (*p* = 0.382).

### 3.2. Bladder Domain

A total of 104/306 (34%) participants reported bladder dysfunction with overall rates equally distributed in the 1st, 2nd, and 3rd trimesters of pregnancy (*p* = 0.168). Symptoms related to urinary distress were more intense in the 3rd trimester of pregnancy compared to the 1st and 2nd trimesters of pregnancy. As seen in Table 3, total bother scores distinguished a significant difference between the 3rd trimester [1.46 (1.04–2.3)] and the 2nd [1.25 (0.62–1.87)] and 1st trimester [1.15 (0.67–1.82)] with no significant difference between the 1st and 2nd trimesters, (*p* = 0.004).

Significantly higher rates of nocturia, the need for pads due to incontinence, and a false sensation of UTI (presented by answers to bladder question 2, question 11, and question 14, respectively) were recorded in the 3rd trimester compared to the 1st and 2nd trimesters (Table 4).

### 3.3. Bowel Domain

A total of 112/306 (36.3%) participants reported intestinal dysfunction with overall rates equally distributed in the 1st, 2nd, and 3rd trimesters of pregnancy (*p* = 0.691). Bother scores calculated for 1st [1.61 (0.64–2.5)], 2nd [1.29 (0.97–2.25)], and 3rd trimesters [1.61 (0.97–2.25)] were compared, with no significant differences observed between the three trimesters (*p* = 0.709).

### 3.4. Prolapse Domain

Thirty-three women (33/306; 10.8%) reported prolapse symptoms with overall rates equally distributed in the 1st, 2nd^,^ and 3rd trimesters of pregnancy. However, when the bother scores were computed, a significant difference between the scores calculated for the 3rd trimester [0.65 (0–1.29)] and those calculated for the 2nd [0.38 (0–0.94)] and 1st trimesters [0.24 (0–0.8) was noted (*p* = 0.012) with no significant difference between the 1st and 2nd trimesters. A significantly higher rate of bulging symptoms either at rest or during effort (presented by answers to prolapse question 2 and prolapse question 3, respectively) was recorded in the 3rd trimester compared to the 1st and 2nd trimesters of pregnancy (Table 4).

### 3.5. Pelvic Floor Total Burden

There were no significant differences in prevalence between the groups when the domain items were counted in total. When the bother scores of all four domains were summed, a significant difference between the scores calculated for the 3rd trimester [5.31 (3.62–7.12)] and those calculated for the 2nd [4.41 (3.14–5.98)] and 1st trimester [3.99 (2.87–5.44)] was noted (*p* = 0.005), with no significant difference between the 1st and 2nd trimesters of pregnancy.

### 3.6. Risk Factors

Five related risk factors were explored: age over 35 years, familial predisposition, body mass index > 25 kg/m^2^, cigarette smoking, and a subjective inability to voluntarily contract the pelvic floor muscles. Overall, 69.6% of the participants demonstrated either one (48.4%), two (17.3%), or three (3.9%) risk factors that were equally distributed among the groups.

## 4. Discussion

Pelvic floor distress symptoms have been sufficiently studied in the third trimester of pregnancy whereas their presence in early pregnancy has been sparsely investigated. To the best of our knowledge, this is the first study that investigates the frequency and assortment of pelvic floor dysfunction complaints throughout pregnancy and examines the degree of bother they cause using a valid questionnaire for pregnant women. To define the prevalence and severity of PFD symptoms all along, rather than in the 3rd trimester of pregnancy, we employed a self-completed, validated, and reliable tool created to assess all four domains of pelvic floor function (bladder, bowel, prolapse, and sexual function) [13]. Pelvic floor outcomes, including stress incontinence, anal incontinence, prolapse, and sexual dysfunction were measured at different stages of pregnancy using the PFQPP. We found that 60.8% of the study cohort endured at least one PFD symptom during pregnancy and that specific PFD symptoms intensified in the third trimester of pregnancy.

Previous studies have underlined that sexual dysfunction is extremely common in the third trimester of pregnancy, with it overwhelmingly impacting women’s quality of life [14,15]. Several studies report a significant decline in sexual desire and sexual activity during the third trimester of pregnancy [14,15,16,17]. Sexual dysfunction, in the 3rd trimester, is mediated by a lack of adequate information about sex in pregnancy and the worry that intercourse may harm the pregnancy by inducing preterm labor or premature rupture of membranes. A recent survey using the Italian PFQPP revealed that in the third trimester of pregnancy, one-third of women abstained from sexual activity and half of them had dyspareunia [14]. The decline in sexual activity is also due to a lack of sexual desire [14]. Sexual dysfunction can lead to depression, anxiety, hypervigilance to pain, negative body image, and low self-esteem and is strongly correlated with urinary incontinence [18]. Our study discloses that sexual inactivity and/or dysfunction, reported by as many as 39.2% of participants, were equally dispersed in the first, second, and third trimesters of pregnancy and did not intensify in the third trimester denoting that sexual dysfunction is promoted by the pregnancy condition itself.

A trend towards Increased rates (5.6%, 7.2%, and 13.4% in the 1st, 2nd, and 3rd trimesters, respectively) of prolapse symptoms did not reach significance probably due to the small number of women experiencing prolapse symptoms (*n* = 33; 10.8%). However, when the bother scores were computed, a significant difference between the scores calculated for the 3rd trimester and those calculated in the 1st and 2nd trimesters was obtained denoting that bulging symptoms were more disturbing in the 3rd trimester of pregnancy (Table 4). Urinary complaints reported by as many as 34% of participants showed significantly higher rates for nocturia and the need to use pads in the 3rd trimester compared to the 1st and 2nd trimesters which is in line with previous observations that hold that the bother instigated by urinary tract symptoms is most frequent in the 3rd trimester [19]. The rate and magnitude of bowel complaints, reported by 36.3% of participants, were similar in the three groups.

Well-established risk factors for PFD, although expected to be more salient among women experiencing PFD symptoms in early pregnancy, were found to be equally distributed within the three groups (Table 1). Smokers were scarce in the cohort as a whole (*n* = 11; 4.6%).

Pelvic floor rehabilitation for pregnant women, displaying PFD symptoms, is hindered by underdiagnosis, embarrassment, fear of stigma, high costs, long waiting periods, or restricted access to medical services. PFD symptoms even though over-expressed during pregnancy remain at large underdiagnosed and undertreated. Pregnant women tend to regard PFD symptoms as a common and transient discomfort associated with pregnancy and do not consider themselves liable to still suffer from PFD symptoms later in life and generally do not seek medical attention. Urinary incontinence and the sensation of pelvic organs bulging through the vagina impede daily functions and depress self-image. These distressing symptoms certainly merit medical attention, sound counselling, and prompt management. A care plan for PFD prevention or treatment during pregnancy is lacking and the need to increase women’s knowledge of PFD and motivate them to engage in PFD prevention is delayed by time constraints during prenatal visits. Women should be intentionally informed and counseled about PFD symptoms so that those who are overweight, smokers, or unable to actively contract their pelvic floor muscles can receive targeted counseling about their increased risk and take preventive action [20].

Intervention strategies that can reduce and lessen PFD symptoms include weight reduction, participation in sports, and pelvic floor muscle training (PFMT) [21]. PFMT is a conservative intervention that can improve bladder, bowel, prolapse, and sexual function. PFMT proven effective to treat stress and urge incontinence [21,22] and employed before or as of early pregnancy is expected to prevent or alleviate PFD symptoms. It has been reported that PFMT improves sexual function in postmenopausal women and when introduced postnatally [23,24]. However, studies describing the effect of PFMT on sexual function and other PFD symptoms during pregnancy are lacking. Obviously, such studies are difficult to achieve since most PFD symptoms are rather mild and remain underreported in early pregnancy. Pelvic floor muscle training which is devoid of serious adverse effects and has been recommended as a first-line treatment in the general population may prove essential to pregnant women.

Lately, eHealth (digital health) intervention programs have been mounted to provide information about pelvic floor ailments and management. A thorough meta-analysis conducted by Xu et al. [25] shows that eHealth intervention is useful in recovering PFD symptoms, especially as regards to outcomes such as stress and urinary incontinence, quality of life, pelvic floor muscle strength, and sexual function. Due to their anonymity, flexibility, and accessibility, eHealth interventions can reduce women’s sense of embarrassment, reduce cost and time, and simplify access to healthcare services. eHealth intervention programs should be made accessible to women during pregnancy and after birth. Treatment of PDF following delivery remains the principle considering that two-thirds of women are still bothered by PFD symptoms one year after delivery [26]. PFD symptoms which are not considered life-threatening overwhelmingly disturb women’s life and are associated with decreased body image and postpartum depression [27,28,29,30].

Our results show that marginal PFD symptoms emerge early in pregnancy and intensify throughout pregnancy. It remains mandatory to screen pregnant women early in pregnancy and to offer them sound counseling and treatment.

### Strengths and Limitations

Anonymous questionnaires, especially in sensitive areas such as sexual incompetence, are more likely to uncover symptoms that might otherwise remain hidden. Anonymity, however, hampers the assessment of potential influencers and confounders not investigated by the questionnaire itself. We acknowledge the need for a longitudinal study that queries and assesses responses from the same participant at different stages of pregnancy. This again may be hindered by anonymity. The PFQPP has been fundamental in the early diagnosis of PFDs in pregnancy, and, to the best of our knowledge, this is the first study to examine differences between trimesters using a valid pregnancy-targeted questionnaire.

## 5. Conclusions

Sixty percent of the participants suffered from at least one pelvic floor disorder. Our study shows that bladder and prolapse symptoms are frequent throughout pregnancy but significantly intensify in the 3rd trimester. Sexual dysfunction remains equally distressing throughout pregnancy, thereby mearing quality of life and self-esteem. Bowel dysfunction is less prevalent compared to other PDF symptoms and the recorded trend towards increased severity during pregnancy that did not reach significance merit further investigation. The preponderance of PDF symptoms, although subtle in early pregnancy, requires that pregnant women be aware of such symptoms early. Healthcare professionals need to adopt a proactive approach and motivate pregnant women to indulge in PFMT. PFQPP emerges as a precious, easy use, risk estimate tool that awards early detection of PDF.

## Figures and Tables

**Table 1 healthcare-11-01096-t001:** General characteristics and risk factors in pregnancy trimesters. PFC = pelvic floor contraction; PFDs = pelvic floor dysfunction symptoms.

Characteristics and Risk Factors	All Patients *n* = 306	1st Trimester *n* = 36	2nd Trimester *n* = 83	3rd Trimester *n* = 187	*p*
Maternal Age	32.6 (±4.6)	33.4 (±4)	32.8 (±4.7)	32.4 (±4.8)	0.481
Gestational Age	30.5 (±20)	10.25 (±1.7)	19.9 (±4.4)	34.8 (±3.6)	<0.001 ^b^
Pregestational weight (kg)	62 (±12.4)	61.1 (±11)	61.8 (±12)	62.8 (±12)	0.607
Current weight (kg)	70 (±12.8)	62.2 (±11)	67.1 (±12)	72.7 (±12)	<0.010 ^a^
Age > 35	75 (25.9%)	11 (31.4%)	19 (25%)	45 (25.1%)	0.725
Familiarity with PFDs (*n* = 304)	34 (11.2%)	3 (8.6%)	14 (16.9%)	17 (9.1%)	0.156
Smoking (*n* = 306)	14 (4.6%)	2 (5.6%)	1 (1.2%)	11 (5.9%)	0.206
Pelvic floor contraction inability (*n* = 304)	15 (4.9%)	2 (5.6%)	5 (6.1%)	8 (4.3%)	0.671

Age at interview was not provided by 16 participants. ^a^ Denotes a significant difference between all three trimesters. ^b^ Denotes a significant difference between the 3rd trimester and either the 2nd or 1st trimester, with no significant difference calculated between the 1st and 2nd trimesters.

**Table 2 healthcare-11-01096-t002:** Prevalence of PFD symptoms in the three pregnancy trimesters.

PFD Symptoms (*n* = 306)	All Trimesters	1st Trimester (*n* = 36)	2nd Trimester (*n* = 83)	3rd Trimester (*n* = 187)	*p*
No symptoms	120 (39.2%)	19 (52.7%)	35 (42.1%)	66 (35.3%)	0.117
Vesical Dysfunction	104 (34%)	9 (25%)	24 (28.9%)	71 (38%)	0.716
Bowel Dysfunction	112 (36.3%)	11 (30.6%)	30 (36.1%)	71 (38%)	0.691
Pelvic Support Dysfunction	33 (10.8%)	2 (5.6%)	6 (7.2%)	25 (13.4%)	0.182
Sexual Dysfunction	65 (21.2%)	6 (16.7%)	16 (19.3%)	43 (23%)	0.611
Sexual Inactivity	55 (18.4%)	3 (8.3%)	15 (18.5%)	37 (20%)	0.352

**Table 3 healthcare-11-01096-t003:** Bother scores calculated for each trimester separately.

Bother Scores	1st Trimester	2nd Trimester	3rd Trimester	*p*
Urinary domain score	1.15 (0.67–1.82)	1.25 (0.62–1.87)	1.46 (1.04–2.3)	0.004 ^b^
Intestinal domain score	1.61 (0.64–2.5)	1.29 (0.97–2.25)	1.61 (0.97–2.25)	0.709
Pelvic domain score	0.24 (0–0.8)	0.38 (0–0.94)	0.65 (0–1.29)	0.012 ^b^
Sexual domain score	1.04 (0.41–2.08)	1.25 (0.41–2.08)	1.25 (0.42–2.5)	0.382
TOTAL score	3.99 (2.87–5.44)	4.41 (3.14–5.98)	5.31 (3.62–7.12)	0.005 ^b^

^b^ Denotes a significant difference between the scores calculated for the 3rd trimester and either the 2nd or 1st trimester, with no significant difference calculated between the 1st and 2nd trimesters.

**Table 4 healthcare-11-01096-t004:** Specific items in the questionnaire that were significant between the trimesters.

Significant PFQPP Items	1st Trimester	2nd Trimester	3rd Trimester	*p*
V2 Presence of nocturia	16 (44.4%)	36 (43.9%)	119 (63.6%)	0.004 ^b^
V11 Use of pads due to urinary leakage	5 (13.9%)	13 (15.7%)	53 (28.3%)	0.028 ^b^
V14 False sensation of UTI	3 (8.3%)	14 (17.1%)	48 (25.7%)	0.037 ^c^
P2 Sensing prolapse of uterus or vagina at rest	2 (5.6%)	11 (13.3%)	40 (21.5%)	0.035 ^c^
P3 Sensing prolapse during physical effort.	2 (5.6%)	15 (18.1%)	49 (26.3%)	0.014 ^c^

^b^ Denotes a significant difference between the 3rd trimester and either the 2nd and 1st trimester, with no significant difference calculated between the 1st and 2nd trimesters. ^c^ Denotes a significant difference only between the 3rd trimester and 1st trimester.

## Data Availability

Data supporting the reported results can be found in a specific archived database generated during the study and they are available upon reasonable request.

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
