# Peer review of "Prevalence and Severity of Pelvic Floor Disorders during Pregnancy: Does the Trimester Make a Difference?"

_healthcare, 2023, doi:10.3390/healthcare11081096_

Round 1

Reviewer 1 Report

The authors present a study of PFD in pregnant women in all three trimesters of pregnancy. The study is interesting, but I have detected some areas for improvement 

Abstract, line 14, symptoms and prevalence of what? 

Line 58, add meaning of QoL. 

Line 67 "publicized risk factors" I think there is a translation error. 

Sometimes they say PFD and sometimes PDF, I understand that it refers to the same thing so I think they should be consistent in these abbreviations. 

I think they should point out the research gap and then indicate the objectives of their work more clearly.

In the methodology, the objective appears in lines 83 and 84, it should be before this section.

How was the data collection carried out and where were the women recruited?

I understand from your wording that the PFQPP questionnaire was used but you only used a certain selection of items? 

Sometimes p's are represented to two decimal places, as in the tables, sometimes to one (line 154) and sometimes to three (line 158). This should always follow the same pattern. 

In table 4 the columns do not have a title. 

The results are a bit short, there are certain things that due to the gestation process itself will be altered. However, I think that the risk factors can be something that can provide much more information in the wording of these results, are there differences in pelvic floor problems between smokers and non-smokers, between people with a higher or lower BMI, between people with a higher or lower BMI?  To complement what has been obtained. The ideal would be to carry out a longitudinal study to check how these changes in the pelvic floor are produced during pregnancy. 

I think there are aspects of the discussion that could benefit from an improvement in the wording and a greater comparison with other studies.  

The conclusions are poorly elaborated.

Author Response

Dear Reviewer,

Response to Reviewer 1 Comments

Comment: Abstract, line 14, symptoms and prevalence of what?

Response: Provided. 

Comment: Line 58, add meaning of QoL.

Response: Added as requested (line 50)

Comment: Line 67 "publicized risk factors" I think there is a translation error. Response: Replaced by "ascertained".

Comment: Sometimes they say PFD and sometimes PDF, I understand that it refers to the same thing so I think they should be consistent in these abbreviations.      

Response: PDF replaced by PFD wherever needed.

Comment: I think they should point out the research gap and then indicate the objectives of their work more clearly.

Response: We complied with the suggested. Provided at the end of paragraph 1 as such: "Pelvic floor distress symptoms have been adequately investigated the third trimester of pregnancy whereas their presence in early pregnancy was rather overlooked. We hypothesized that pelvic symptoms remain under reported in early and mid- pregnancy until distress worsens the third trimester of pregnancy. Our objective was to delineate variances in PFD symptomatology between the three trimesters of pregnancy. We thus compared the prevalence and severity of PFD symptoms in a subset of women who have anonymously completed the Italian PFQPP at different stages of pregnancy."

Comment: In the methodology, the objective appears in lines 83 and 84, it should be before this section.

Response: Now incorporated in the last paragraph of the introduction.  

Comment: How was the data collection carried out and where were the women recruited?

Response: Women were recruited at - outpatient pregnancy follow up clinics – of two tertiary medical centers namely, Santi Paolo e Carlo, San Paolo Hospital, Milan, Italy, and San Gerardo Hospital, Monza, Italy – now added to the text lines 90-91 and 97.  

Comment: I understand from your wording that the PFQPP questionnaire was used but you only used a certain selection of items?

Response: Participants answered all the items in PFQPP (the rate of missing items was 1%).

Comment: Sometimes p's are represented to two decimal places, as in the tables, sometimes to one (line 154) and sometimes to three (line 158). This should always follow the same pattern. 

Response: all p values are now provided with three decimals.

Comment: In table 4 the columns do not have a title.                                    

Response: added.

Comment: The results are a bit short, there are certain things that due to the gestation process itself will be altered. However, I think that the risk factors can be something that can provide much more information in the wording of these results, are there differences in pelvic floor problems between smokers and non-smokers, between people with a higher or lower BMI, between people with a higher or lower BMI?  To complement what has been obtained. The ideal would be to carry out a longitudinal study to check how these changes in the pelvic floor are produced during pregnancy.

Response: We agree that a longitudinal study is ideal in order to complement our observations. This has been recognized in the section addressing the limitations of the study and worded as such "We acknowledge the need for a longitudinal study that queries and assesses responses from the same participant at different stages of pregnancy. This again may be hindered by anonymity". Risk factors were equally distributed within the three subgroups. The differences in BMIs along pregnancy is due to the pregnancy itself and did not award space for comparison between groups.  Smoking was acknowledged by mere 14 (4.6%) of participants (2, 1 and 11 in the 1st, 2cd and third trimesters respectively) – now specified in the text. Studying the risk factors within each group separately unrelated to the trimester of pregnancy is hindered by the small number of participants in the first and second trimesters (n = 36 and n = 83 respectively). The contribution of risk factors to PDF symptoms was indeed confirmed by co-authors Palmieri et al. (reference no. 8) and worded as such "Familiarity for PFDs, pelvic floor contraction inability, cigarette smoking, body mass index more than 25 (calculated as weight in kilograms divided by the square of height in meters), and age more than 35 years were confirmed risk factors".

Comment: I think there are aspects of the discussion that could benefit from an improvement in the wording and a greater comparison with other studies.

Response: THANKS for the suggestion. The reamended version hopefully answers these worries.

Comment: The conclusions are poorly elaborated.

Response: The conclusions were rephrased. Hopefully better presented. Thanks for the suggestion.

For more details please see the revised version manuscript.

Reviewer 2 Report

METHODS

Line 81: I suppose ref 4 should be ref 7.

Selection of women is not clear. If ref 7 describes the original study population that was used for the current secondary analysis, then 2007 women could have been analyzed. How did you select the current 306 women? From the original study recruitment is not completely clear – is this a cohort of women visiting the antenatal clinic, or a selection based on unspecified criteria?

You might consider to add an English translation of the Italian PFQPP (or the German original version) as an addendum.

RESULTS

Table 1: What is “Pelvic floor contraction inability” this is not defined in your methods. I suppose the figures behind the row captions denote that there are some missing cases. If you miss 16 case for age > 35 the you also miss 16 cases for average age. You could specify this below the table. I suppose you did a 2 x 3 chi square test, or did you do 3 pairwise comparisons, please specify. Ref b is not explained below the table. As you how p values it is superfluous to specify that a difference is statistically significant by ref a.

Table 2: You might make top row in the table specifying the number of women without any symptoms, instead of a remark below the table.

Table 3: It is not clear how you calculate the “bother score” – is this an average for the complete group, although part of the women have no complaints at all?

Table 4: Column captions are missing (as in other tables).

DISCUSSION

You remark that previous studies have underlined that sexual dysfunction is extremely common in the third trimester of pregnancy. How does this compare to your findings of 20% in the 3d trimester?

Author Response

Dear Reviewer,

Response to Reviewer 2 Comments

Comment: Line 81: ref 4 should be ref 7.

Response: Yes indeed. Thanks. Corrected. Now reference 8.

Comment Selection of women is not clear. If ref 7 describes the original study population that was used for the current secondary analysis, then 2007 women could have been analyzed. How did you select the current 306 women? From the original study recruitment is not completely clear – is this a cohort of women visiting the antenatal clinic, or a selection based on unspecified criteria?

Response: The original study included indeed 2007 participants of whom 1048 were pregnant and 959 were postpartum women. Pregnant women were recruited at eight tertiary medical centers. Of these, only two centers (Santi Paolo e Carlo, San Paolo Hospital, Milan, Italy, and San Gerardo Hospital, Monza, Italy) had questionnaires completed in early and mid-pregnancy, restricting unfortunately the number of participants to 306. The cohort of women was recruited at antenatal clinics. This information is now clarified in the methodology section.

Comment: You might consider adding an English translation of the Italian PFQPP (or the German original version) as an addendum.

Response: We added the German original version as an addendum.

Comment: Table 1: What is “Pelvic floor contraction inability” this is not defined in your methods. I suppose the figures behind the row captions denote that there are some missing cases. If you miss 16 case for age > 35 the you also miss 16 cases for average age. You could specify this below the table. I suppose you did a 2 x 3 chi square test, or did you do 3 pairwise comparisons, please specify. Ref b is not explained below the table. As you how p values it is superfluous to specify that a difference is statistically significant by ref a.

Response: (a) “Pelvic floor contraction (PFC) inability” refers to the "subjective ability to voluntarily contract the pelvic floor muscles at the interview time point" (Metz et al. ref no. 10). The question posed "are you capable to contract the muscles in the pelvic floor?" requires one of three contributary answers re: yes, i do not know and no. The term “Pelvic floor contraction inability” is the term used by our co-authors in MS no. 8 and has been here used for sake of consistency.

(b) The explanation for symbol b was added to the footnote.

(c) Unfortunately, we could not retrieve the age of 16 participants. Now specified below the table.

(d) Chi square was used for comparison of categorial Table 1 depicts categorical variables, in a 2X3 table) if a significant difference was found between trimesters, a comparison was made between each pair.

Comment: Table 2: You might make top row in the table specifying the number of women without any symptoms, instead of a remark below the table.

Response: We did. Thanks for the suggestion.

Comment: Table 3: It is not clear how you calculate the “bother score” – is this an average for the complete group, although part of the women have no complaints at all?                      

Response: Yes, it is indeed the average of the whole group. Bother is scored on a 5-point Likert Scale with the following choice of answers: "1 - not at all”; “2 - a little”; “3 - quite a lot” and “4 - very much”.  This information is provided in the methodology section.

Comment: Table 4: Column captions are missing (as in other tables).

Response: Completed as requested.  

Comment: You remark that previous studies have underlined that sexual dysfunction is extremely common in the third trimester of pregnancy. How does this compare to your findings of 20% in the 3d trimester?

Response: Our results show that 132 (40.4%) of participants reported sexual inactivity and/or sexual dysfunction. Sexual dysfunction is mostly due to dyspareunia. These numbers are a bit inferior to those reported by co-authors Frigerio (quoted ref 13) but in line with those displayed by others.

For more details please see the revised version manuscript.

Round 2

Reviewer 1 Report

The authors have implemented the suggested modifications and improvements. I consider the final work to be of higher quality, at this stage I can detect no further comments or improvements to be implemented.